# Combined Inhibition of TGF-β1-Induced EMT and PD-L1 Silencing Re-Sensitizes Hepatocellular Carcinoma to Sorafenib Treatment

**DOI:** 10.3390/jcm10091889

**Published:** 2021-04-27

**Authors:** Ritu Shrestha, Prashanth Prithviraj, Kim R. Bridle, Darrell H. G. Crawford, Aparna Jayachandran

**Affiliations:** 1Faculty of Medicine, The University of Queensland, Brisbane, QLD 4120, Australia; ritu.shrestha@uq.edu.au (R.S.); k.bridle@uq.edu.au (K.R.B.); d.crawford@uq.edu.au (D.H.G.C.); 2Gallipoli Medical Research Institute, Greenslopes Private Hospital, Brisbane, QLD 4120, Australia; 3Fiona Elsey Cancer Research Institute, Ballarat, VIC 3350, Australia; Prashanth@BallaratOncology.com.au; 4School of Science, Psychology and Sports, Federation University Australia, Ballarat, VIC 3350, Australia

**Keywords:** hepatocellular carcinoma, EMT, immune checkpoint, sorafenib, PD-L1

## Abstract

Hepatocellular carcinoma (HCC) is the most common type of primary hepatic malignancy. HCC is one of the leading causes of cancer deaths worldwide. The oral multi-tyrosine kinase inhibitor Sorafenib is the standard first-line therapy in patients with advanced unresectable HCC. Despite the significant survival benefit in HCC patients post treatment with Sorafenib, many patients had progressive disease as a result of acquiring drug resistance. Circumventing resistance to Sorafenib by exploring and targeting possible molecular mechanisms and pathways is an area of active investigation worldwide. Epithelial-to-mesenchymal transition (EMT) is a cellular process allowing epithelial cells to assume mesenchymal traits. HCC tumour cells undergo EMT to become immune evasive and develop resistance to Sorafenib treatment. Immune checkpoint molecules control immune escape in many tumours, including HCC. The aim of this study is to investigate whether combined inhibition of EMT and immune checkpoints can re-sensitise HCC to Sorafenib treatment. Post treatment with Sorafenib, HCC cells PLC/PRF/5 and Hep3B were monitored for induction of EMT and immune checkpoint molecules using quantitative reverse transcriptase (qRT)- PCR, western blot, immunofluorescence, and motility assays. The effect of combination treatment with SB431542, a specific inhibitor of the transforming growth factor (TGF)-*β* receptor kinase, and siRNA mediated knockdown of programmed cell death protein ligand-1 (PD-L1) on Sorafenib resistance was examined using a cell viability assay. We found that three days of Sorafenib treatment activated EMT with overexpression of TGF-*β*1 in both HCC cell lines. Following Sorafenib exposure, increase in the expression of PD-L1 and other immune checkpoints was observed. SB431542 blocked the TGF-*β*1-mediated EMT in HCC cells and also repressed PD-L1 expression. Likewise, knockdown of PD-L1 inhibited EMT. Moreover, the sensitivity of HCC cells to Sorafenib was enhanced by combining a blockade of EMT with SB431542 and knockdown of PD-L1 expression. Sorafenib-induced motility was attenuated with the combined treatment of SB431542 and PD-L1 knockdown. Our findings indicate that treatment with Sorafenib induces EMT and expression of immune checkpoint molecules, which contributes to Sorafenib resistance in HCC cells. Thus, the combination treatment strategy of inhibiting EMT and immune checkpoint molecules can re-sensitise HCC cells to Sorafenib.

## 1. Introduction

Hepatocellular carcinoma (HCC) is the most common form of primary liver malignancy. Globally, HCC is the fourth leading cause of cancer-related deaths, with approximately 782,000 deaths in 2018 [1,2]. The worldwide health burden of this disease is increasing with minimal survival rates and limited therapeutic alternatives [3].

Sorafenib, an oral multikinase inhibitor, was approved by the Food and Drug Administration (FDA) as the first-line treatment in patients with advanced unresectable HCC [4,5]. Although Sorafenib is the standard treatment modality in HCC, the low survival benefit of three months and acquisition of drug resistance along with adverse effects limits its therapeutic benefits [4,6,7,8,9]. Exploring the process of Sorafenib resistance and identifying targets to overcome Sorafenib resistance remains a pressing area of need. Several studies have proposed multiple cellular mechanisms and signaling pathways that result in the development of Sorafenib resistance in HCC, including epithelial-to-mesenchymal transition, cancer stem cells, hypoxia, c-Jun, EGFR and PI3/AKT activation, apoptosis resistance, and others [5,10,11,12,13,14,15,16,17]. Studies have also suggested several combination therapies with Sorafenib to improve the therapeutic effectiveness for the treatment of HCC patients [18,19,20,21].

Besides Sorafenib, the FDA has approved other drugs either alone or in combination therapy for HCC patients. Lenvatinib is another FDA-approved first-line treatment for patients with HCC, whereas Regorafenib, Ramucirab, and Cabozantinib are second-line therapies in HCC patients following Sorafenib treatment. Immunotherapeutic treatment approaches based on immune checkpoint inhibition are increasingly being investigated in HCC, resulting in the approval of immune checkpoint inhibitors (ICIs) against programmed cell death protein-1 (PD-1) such as Nivolumab and Pembrolizumab [22,23]. In addition, the FDA has approved the combined treatment of Nivolumab and Ipilimumab as another second-line therapy for HCC patients previously treated with Sorafenib. Recently, the FDA has approved the combination ICI against PD-L1, Atezolizumab with Bevacizumab as a first-line treatment for HCC patients based on the response rates in the IMbrave150 trial [24]. Aberrant expression of immune checkpoints including PD-L1 in HCC patients is associated with significantly worse clinical outcomes [25,26].

Epithelial-to-mesenchymal transition (EMT) is a multistep biological process regulating the acquisition of mesenchymal phenotype and function by epithelial cells. Several studies have implicated the process of EMT in cancer initiation and progression [5,27]. The process of EMT enables tumour cells to migrate to secondary sites, whereas the reverse process of mesenchymal-to-epithelial transition (MET) allows tumour cells to settle, proliferate, and differentiate, forming secondary tumours [27,28,29]. EMT is a crucial process utilized by HCC cells to acquire resistance to Sorafenib treatment [5,30,31,32]. In HCC patients, the process of EMT coincides with immune checkpoint expression such as PD-L1, resulting in poor prognosis for patients with these tumour characteristics [26,33].

Finding additional druggable targets for Sorafenib resistant HCC cells could greatly enhance the chances for the evolution of efficacious combination therapies. Thus, the aim of the present research is to facilitate the identification of novel combination therapies by exploring the relationship between EMT, immune checkpoints, and Sorafenib resistance in HCC. Changes in EMT features and immune checkpoint expression were monitored following Sorafenib treatment of HCC cells. We then explored the potential of blocking both EMT and immune checkpoint molecules to attenuate Sorafenib resistance in HCC.

## 2. Materials and Methods

### 2.1. Cell Culture

As previously reported, both human HCC cell lines, Hep3B (obtained from Prof. V. Nathan Subramaniam, The Queensland University of Technology, Queensland, Australia) and PLC/PRF/5 (procured from CellBank Australia (85061113)), were mycoplasma-tested with the MycoAlert test (Abm, Richmond, BC, Canada), as previously described [33]. Dulbecco’s modified Eagle’s medium (DMEM) (Thermofisher, Victoria, Australia), comprising 10% fetal bovine serum (FBS) (Gibco, Victoria, Australia) and 1% penicillin/streptomycin (P/S) (Thermofisher, Victoria, Australia), was used to culture both cell lines, as previously described [34].

### 2.2. Reagents

Sorafenib was procured from Selleckchem.com, Australia, and reconstituted as per the manufacturer’s instructions. The HCC cells were exposed to Sorafenib for three days at concentrations ranging from 0–12.5 µM. To induce EMT, the HCC cells were exposed for three days with 10 ng/mL TGF-*β*1 (Peprotech, Lonza, Victoria, Australia). TGF-*β*1 was reconstituted in 10 mM citric acid, pH 3.0, as per the manufacturer’s protocol. SB431542, a TGF-β receptor kinase inhibitor (Sigma, New South Wales, Australia), was used at a concentration of 2 µM. SB431542 was dissolved in DMSO as per the manufacturer’s protocol.

### 2.3. RNA Extraction and cDNA Synthesis

As previously described, RNA was purified and quantified with Isolate II Bioline RNA synthesis kit (Bioline, New South Wales, Australia) and Nanodrop 2000c (Thermofisher, Victoria, Australia), respectively [34]. Bioline SensiFAST cDNA synthesis kit (Bioline, New South Wales, Australia) was used to reverse transcribe 1 µg RNA to cDNA.

### 2.4. Quantitative Reverse Transcription-PCR (qRT-PCR)

qRT-PCR was performed on a ViiA7 Applied Biosystems Real-Time PCR system with Lo-ROX SYBR Green (Bioline, New South Wales, Australia) [34]. Briefly, a three-step cycle protocol was performed with 40 cycles of the following set-up with temperatures of 95, 63, and 75 °C for 5, 20, and 20 s, respectively. The internal control used was Beta-Actin (*ActB*). The primers used were previously reported [33] and listed in Table 1. Expression levels are shown as copies of the selected gene per 10,000 copies of *ActB*. The 2ΔΔCt method was used for data analysis, wherein selected gene expression was normalised to *ActB* expression. Data are represented as copies of selected gene per 10,000 copies of *ActB*.

### 2.5. Western Blot Analysis

Cells were seeded and treated in six-well plates. Total proteins were extracted from the cells using a RIPA buffer (Thermofisher, Victoria, Australia) with Complete (Roche, New South Wales, Australia) and phosSTOP (Roche, New South Wales, Australia) protease and phosphatase inhibitors at 4 °C. The total protein was measured with a Pierce BCA protein assay kit (Thermofisher, Victoria, Australia); 10 µg of protein was used for separation by electrophoresis (SDS-PAGE) in a polyacrylamide gel in the presence of sodium dodecyl sulphate (SDS) and transferred to a polyvinylidene difluoride film (PVDF) membrane. The membranes were blocked with 5% skim milk in Tris-buffered saline containing 0.1% Tween 20 (TBS-T) and incubated overnight with primary antibodies at 4 °C. The membranes were then incubated with HRP-conjugated secondary antibodies and proteins detected by SuperSignal West Femto Maximum Sensitivity Substrate (Thermofisher, Victoria, Australia). Glyceraldehyde 3-phosphate dehydrogenase (GAPDH) or βActin was used as the housekeeping control. The images were captured and quantified with Image Quant LAS 500 and Image Studio Lite Ver 5.2 software, respectively. Antibodies used are listed in Table 2.

### 2.6. Immunofluorescence

Cells were plated at a density of 1 × 10^4^ per chamber in eight-well tissue culture treated chamber slides. After 1XPBS wash, cells were treated with 4% paraformaldehyde (Fisher scientific, Victoria, Australia) and 0.1% Triton X-100 (Sigma-Aldrich, New South Wales, Australia). This was followed by blocking with 5% FBS and incubation with primary antibodies overnight at 4 °C. After incubation with secondary antibodies at room temperature for 30 min, cells were incubated with 4′,6-diamidino-2-phenylindole (DAPI) (Thermofisher Scientific, Victoria, Australia) for 10 min at room temperature. ProLong Diamond (Thermofisher Scientific, Victoria, Australia) was used to mount the slides. Nikon C2 system and NIS-Elements software (Nikon, Australia) were used to capture and analyze the images. Antibodies used were previously reported [33] and listed in Table 2.

### 2.7. Wound Healing Assay

Cells were seeded at density of 5 × 10^4^ per well into a 24-well plate. After cells reached confluency, a scratch or wound was made using a sterile pipette tip and washed with 1XPBS. Photographs were taken at the indicated time points with an inverted Olympus DP21 microscope (Olympus, Tokyo, Japan). The relative wound closure was quantified with the Fiji plug-in for Image J software version 1.53c (National Institutes of Health, Bethesda, MD, USA).

### 2.8. Transwell Migration Assay

First, 1 × 10^5^ cells in a serum free DMEM culture medium were seeded into the top chamber of 8-µm pore sized Transwell chambers (Corning, Rowe Scientific, Queensland, Australia). The bottom chamber was filled with 500 µL of DMEM culture medium with 10% FBS to serve as the chemoattractant. Cells were incubated for 24 h at 37 °C, fixed with 4% paraformaldehyde (Fisher Scientific, Victoria, Australia) for 15 min, and stained with 0.1% Crystal Violet (Sigma-Aldrich, New South Wales, Australia). After removing cells on the top chamber, the migrated cells on the bottom chamber were photographed with Olympus DP21. For quantification of cell migration, the Crystal Violet staining on the transwell membrane was extracted using 5% SDS, and a 570 nm wavelength was used to measure absorbance.

### 2.9. Cell Viability Assay

Cells were seeded in 96-well plates at a density of 1 × 10^3^ per well and treated as indicated. Cell proliferation was quantified with the CellTitre 96 Aqueous one solution cell proliferation assay (MTS) from Promega, Australia, according to the manufacturer’s instructions.

### 2.10. PDL-1 Knockdown

Cells were transfected at 50% confluency for transient siRNA transfection. A 10 nM final concentration of a control siRNA (4390843) (Thermofisher, Victoria, Australia) and two distinct silencer select siRNAs against PDL-1 (s26547 and s26548) (Thermofisher, Victoria, Australia) were used with Lipofectamin RNAiMAX (Invitrogen, Victoria, Australia) according to the manufacturer’s instructions. Cells were collected for further experiments 72 h after incubation with the siRNA complex.

### 2.11. Drug Combination Analysis

A web-based application, SynergyFinder 2.0, was utilised to evaluate the synergistic effect of combination drug treatment [35]. The SynergyFinder 2.0 applies algorithms to compute synergy scores for dose–response matrix data. We utilised HSA model for synergy assessment for drug combination in our study [35,36].

### 2.12. Statistical Analysis

All experiments were repeated in biological triplicates, and representative results are presented. Prism software version 8.00 (GraphPad Software Inc.) was used to perform statistical analyses. Student’s *t*-test was used to analyse the differential gene expression between the control and cells treated with Sorafenib. A one-way analysis of variance (ANOVA) followed by Sidak’s multiple comparisons test was used for multiple comparisons. A two-way ANOVA and Tukey’s multiple comparisons test were used for statistical analysis for the cell viability assay. The results are shown as mean ± standard error of mean (SEM). Statistical significance was set at *p* < 0.05. Error bars indicate SEM.

## 3. Results

### 3.1. HCC Cells Undergo EMT upon Sorafenib Exposure

In order to explore the association with EMT and Sorafenib treatment in HCC, PLC/PRF/5 and Hep3B cells were treated with their respective IC_50_ concentration of Sorafenib for 72 h. The IC_50_ concentrations of Sorafenib determined with cell viability assays for PLC/PRF/5 and Hep3B cells were 6.965 and 5.347 µM, respectively (Appendix A). qRT-PCR analysis showed that PLC/PRF/5 cells underwent EMT by reduction in the expression of epithelial makers E-cadherin and Occludin, and concomitant elevation in the expression of mesenchymal markers N-cadherin, Vimentin, Snai1, and Snai2 along with induction of TGF-β1 expression following Sorafenib treatment (Figure 1A). The induction of EMT following Sorafenib treatment in PLC/PRF/5 cells was validated at the protein level (Figure 1B). Given that EMT marker changes are often associated with enhanced motile capability, we demonstrated the migratory potential of PLC/PRF/5 cells was also increased upon Sorafenib treatment, as evaluated by the transwell migration assay (Figure 1C) and wound healing assay (Figure 1D).

Similarly, Sorafenib treatment in Hep3B cells also resulted in the induction of EMT, as demonstrated by qRT-PCR, western blot analysis, transwell migration, and wound healing assays (Appendix A).

However, Sorafenib treatment did not upregulate the tumor necrosis factor—α (TNF-α) expression in either of the cell lines (Appendix A). In addition, we observed that the cells that migrated through the transwell membrane following Sorafenib treatment were positive for proliferation marker ki67, as demonstrated by immunofluorescence staining (Appendix A). This observation confirms that the cells migrating through the transwell membrane are not apoptotic cells, but are cells that have undergone EMT.

### 3.2. Sorafenib Treatment Induces Immune Checkpoint Molecules Expression in HCC Cells

We also evaluated the effects of Sorafenib treatment on immune checkpoint molecule expression in HCC cells. In PLC/PRF/5, Sorafenib treatment resulted in upregulation of immune checkpoints *PD-L1* or *CD274*, *CD73* or *NT5E*, *B7-H3* or *CD276*, *VISTA* or *VSIR*, and *TIM-3* or *HAVCR2*, as demonstrated by qRT-PCR (Figure 2A) and western blot analysis (Figure 2B). Similarly, Hep3B cells displayed elevated expression of *PD-L1*, *B7-H3*, and *VISTA* and decreased expression of *CD73*, as revealed by qRT-PCR (Figure 3C) and western blot (Figure 3D). However, Hep3B cells did not express TIM-3.

### 3.3. SB431542 Inhibits TGF-β1-Mediated EMT in HCC Cells

We have previously observed that TGF-*β*1 treatment induces EMT in HCC cells (manuscript in press). In order to inhibit TGF-*β*1-driven EMT effects in HCC cells, we utilised a TGF-*β* receptor kinases inhibitor, SB431542. We observed that treatment with SB431542 can effectively block TGF-*β*1-mediated EMT effects in PLC/PRF/5 cells, as evidenced by changes in expression of EMT markers, as demonstrated by qRT-PCR (Figure 3A) and western blot (Figure 3B). We also observed a similar blockade of TGF-*β*1-induced EMT effects in Hep3B cells by both qRT-PCR (Figure 3C) and western blot analysis (Figure 3D). Furthermore, the results were validated with fluorescence microscopy in both PLC/PRF/5 and Hep3B cells (Figure 3E).

Furthermore, the functional motility assays revealed that SB431542 can effectively inhibit TGF-β-mediated motility in both HCC cells lines (Figure 4A,B).

### 3.4. SB431542 Attenuates TGF-β1-Induced PD-L1 Expression

In our previous study, we observed that TGF-*β*1 driven EMT coincides with the upregulation of PD-L1expression in HCC cells (Manuscript in press). We thus utilised SB431542 treatment to assess whether blocking TGF-*β*1-induced EMT had any impact on PD-L1 expression. Reduced expression of PD-L1 was observed following SB431542 treatment in PLC/PRF/5 (Figure 5A,B) and Hep3B (Figure 5C,D) cells exposed to TGF-*β*1, as demonstrated by qRT-PCR and western blot analysis. Fluorescence microscopy further confirmed the downregulation of TGF-*β*1-mediated PD-L1 expression following treatment with SB431542 in both PLC/PRF/5 and Hep3B cells (Figure 5E).

### 3.5. Knockdown of PD-L1 in HCC Cells Can Reverse TGF-β1-Mediated EMT

To further confirm the relationship between TGF-*β*1-induced EMT and expression of PD-L1, we knockdown the expression of PD-L1 with two different siRNAs in HCC cells. We confirmed the effective knockdown of PD-L1 with both the siRNAs tested by qRT-PCR and western blot analysis in PLC/PRF/5 (Figure 6A,B). Furthermore, we observed the reversal of TGF-*β*1-induced EMT following PD-L1 knockdown, evidenced by the reversal of both epithelial and mesenchymal marker expression in PLC/PRF/5 cells, as demonstrated by qRT-PCR (Figure 6C), western blot analysis (Figure 6D), and immunofluorescence staining (Figure 6E) after treatment of TGF-*β*1 and PD-L1 siRNA in combination.

Similar reversal of TGF-*β*1-mediated EMT following PD-L1 silencing was also observed in Hep3B cells (Figure 7A–E).

Furthermore, the expression of PD-L1 was also significantly reduced following knockdown, and notably, the levels remained low following treatment with TGF-*β*1 in both PLC/PRF/5 and Hep3B cells (Appendix A).

In addition, the migratory ability of cells was significantly reduced upon knockdown of PD-L1 in both HCC cell lines treated with TGF-*β*1 as demonstrated by transwell migration (Figure 8A) and wound healing assays (Figure 8B).

### 3.6. HCC Cells Can Overcome Sorafenib Resistance by Combined Targeting of PD-L1 and TGF-β1 Signalling

Given that Sorafenib treatment in HCC induces both EMT and immune checkpoint molecules, we hypothesised that EMT and immune checkpoint expression may contribute to the development of Sorafenib resistance in HCC, and thus blocking EMT and the immune checkpoint may re-sensitise Sorafenib resistant cells. In order to confirm the potential synergistic effect of Sorafenib and SB431542 as combination drug treatment in our study, we utilised the web application SynergyFinder 2.0 to assess the synergistic effect of combining Sorafenib and SB431542 based on HSA model. We observed a synergistic effect of Sorafenib and SB431542 with a HSA synergy score of 10.701 (Appendix A).

Next, we combined PD-L1 silencing and SB431542 along with Sorafenib treatment to HCC cells to examine if combined targeting of EMT and PD-L1 can overcome resistance to Sorafenib in HCC cells. At increasing concentrations of Sorafenib, the cell viability assay demonstrated that combination of knockdown of PD-L1 and SB431542 enhanced the cytotoxicity of Sorafenib in HCC cells when compared to treatment with Sorafenib alone or with either SB431542 or PD-L1 siRNA combined with Sorafenib treatment in both HCC cell lines (Figure 9A,B). This confirmed that combined targeting of EMT and PD-L1 has the potential to re-sensitise HCC cells to Sorafenib treatment.

### 3.7. Combining PD-L1 Knockdown with SB431542 Can Reverse Sorafenib-Induced EMT in HCC Cells

To confirm that combining PD-L1 silencing with SB431542 along with Sorafenib can result in less aggressive HCC cells, we assessed the expression of EMT markers and PD-L1 following combination treatment. We observed that the Sorafenib-induced EMT effects on HCC cells are reversed following combination treatment with PD-L1 knockdown and SB431542, as evidenced by qRT-PCR and western blot in PLC/PRF/5 (Figure 10A,B) and Hep3B (Figure 10C,D) cells. These results were further confirmed by fluorescence microscopy in both PLC/PRF/5 and Hep3B cells (Figure 9E).

The expression of PD-L1 also reversed following the combination treatment in both PLC/PRF/5 and Hep3B, as demonstrated by qRT-PCR (Appendix A) and fluorescence microscopy (Appendix A).

The migratory ability of HCC cells also reduced following all combination treatments except with control siRNA and Sorafenib treatment, as demonstrated by the transwell migration assay (Figure 11). Quantification of motility revealed a trend in reduction of motility, although it was not significant for combined treatment of PD-L1 knockdown with SB431542 and Sorafenib compared with control siRNA, SB431542, and Sorafenib treatment in both cell lines. Quantification of motility revealed that PDL1 siRNA was superior to control siRNA in blocking Sorafenib induced motility (Figure 11).

## 4. Discussion

Herein, we report that Sorafenib induces both EMT and expression of immune checkpoints in human HCC cells, PLC/PRF/5 and Hep3B. We used PLC/PRF/5 and Hep3B cells for our study as both these cell lines have previously been utilised as experimental models of Sorafenib resistance [37,38]. We observed increased expression of TGF-*β*1 following Sorafenib treatment. We demonstrated blockade of TGF-*β*1-induced EMT with SB431542, a selective and potent TGF-*β* receptor kinase inhibitor. In addition, we found that siRNA-mediated knockdown of PD-L1 reversed TGF-*β*1-induced EMT. Importantly, we utilised SB431542 and PD-L1 silencing in combination to overcome resistance to Sorafenib in HCC cells. We conclude that the combination of TGF-*β* receptor kinase inhibitor and immune checkpoint inhibitor can synergistically impede EMT and potentially improve the sensitivity of HCC cells to Sorafenib.

The oral multi-tyrosine kinase inhibitor, Sorafenib, is approved as a first-line therapy in patients with advanced unresectable HCC [4]. Despite the significant overall survival benefit of Sorafenib treatment in HCC patients, many patients had progressive disease due to the development of therapeutic resistance [9]. Circumventing resistance to Sorafenib by exploring and targeting possible molecular mechanisms and pathways is an area of active research.

EMT is a crucial mechanism implicated in the development of Sorafenib resistance in HCC [9,16,31]. A study demonstrated that long-term exposure of HCC cell lines, HepG2 and HUH7, and human embryonic liver cells WRL-68 to Sorafenib resulted in development of Sorafenib resistance along with induction of EMT characterised by reduction in E-cadherin and enhanced invasive potential [16]. Likewise, other studies have demonstrated the activation of EMT in Sorafenib-resistant HCC cells with enhanced migratory potential and cell viability [39,40]. These observations are consistent with our results demonstrating that Sorafenib treatment induces drug resistance and EMT in human HCC cell lines with repression of epithelial markers and induction of mesenchymal markers along with increased migratory potential.

In contrast, Chung et al. showed that Sorafenib repressed TGF-*β* responsiveness in HCC cells through degradation of the cell surface TGF-*β*RII [41]. The study reported that Sorafenib blocks TGF-*β*-mediated signalling and cellular responses in TGF-*β*-stimulated hepatoma cells, HepG2, whereas our study reported increased TGF-*β*1 expression in Hep3B and PLC/PRF/5 cells without pre-stimulation with TGF-*β* or similar cytokine. Similarly, Chen et al. also reported inhibitory effects of Sorafenib on the TGF-*β* signalling pathway in mouse hepatocyte cells AML12 stimulated with TGF-β [42]. These differences may be due to the different cell lines utilised in the studies. Furthermore, it is conceivable that Sorafenib may suppress TGF-*β*-mediated effects in pre-stimulated cells, and Sorafenib may induce upregulation of TGF-*β* in non-stimulated hepatoma cells. Given that TGF-*β*1 is a major inducer of EMT and elevated expression is noted after Sorafenib treatment, we hypothesised that TGF-*β*1-induced EMT plays a crucial role in inducing Sorafenib resistance in HCC. An interesting study by Tan W. et al. revealed TNF-α-induced EMT was responsible for the development of resistance to Sorafenib in HCC [43]. Our study is the first to examine association between TGF-*β*1-induced EMT and Sorafenib resistance in HCC.

Immunotherapy based on ICIs has achieved substantial progress and breakthroughs in cancer therapeutics [44]. Immune checkpoint blockade therapy in HCC is a potent therapeutic alternative, as immune activation or suppression determines tumor progression or eradication in HCC [45]. There are several ongoing studies with ICIs in HCC. In particular, there are ongoing clinical studies using combination approaches involving ICIs with either another ICI or other immune and non-immune based treatment modalities [44].

The combination treatment approach with ICIs in HCC can be developed with more efficacy provided there is association between Sorafenib resistance and immune checkpoint molecules. Herein, we report that Sorafenib treatment induces upregulation of immune checkpoints such as *PD-L1*, *CD73*, *B7-H3*, *VISTA*, and *TIM-3* in HCC cells. In line with our study, Lu et al. demonstrated increased expression of PD-L1 in tumor-infliltrating immune cells in 23 Sorafenib-treated patients [46]. Another study utilized the GEO data to report that overexpression of DNA methyltransferases (DNMTs) was associated with upregulation of PD-L1 expression in HCC mice resistant to Sorafenib treatment [40]. In addition, combined treatment of Sorafenib and anti-PD-L1 mAb has resulted in a remarkable decrease in tumour growth in mice [47]. An interesting study by Dong MP et al. reported increased levels of several soluble checkpoints such as BTLA, LAG-3, CTLA-4, and PD-1 after two weeks of exposure to Sorafenib in HCC patients [48].

Studies have reported the association of EMT with immune evasion in tumour cells [27,49]. Furthermore, the EMT score is also known to be associated with the expression of several checkpoint molecules including PD-L1 [50]. The significant relationship between EMT and expression of PD-L1 has been demonstrated in several instances in lung cancer [51,52,53,54]. Similar association of EMT and expression of PD-L1 has been reported in oral squamous cell carcinoma, esophageal cancer, and breast cancer [52,55,56]. In addition, overexpression of PD-L1 is also known to be influenced by TGF-*β*1-mediated immunosuppression and targeting of both PD-L1 and TGF-β with bifunctional protein M7824 inhibits tumour mesenchymalisation and PD-L1 mediated immunosuppression [53]. We have previously reported that immune checkpoint expression is closely correlated with EMT in HCC and is associated with an aggressive clinical course [26,33]. Our previous study confirmed that HCC patients with concomitant higher expression of PD-L1 along with higher expression of mesenchymal marker Vimentin, and lower expression of epithelial marker E-cadherin had poor clinical outcomes [26]. We have identified that EMT induced by either TGF-*β*1 or TNF-*α* regulates immune checkpoint expression in HCC cells [33]. In addition, we found that Sorafenib treatment enhances expression of TGF-*β*1 and not TNF-*α*. Given the close correlation between EMT and immune escape, we hypothesised that TGF-*β*1-induced EMT regulates immune checkpoint expression, thereby driving therapy resistance in HCC cells treated with Sorafenib. To confirm the relationship between TGF-*β*1-induced EMT and expression of immune checkpoint along with Sorafenib resistance, we utilised SB431542, a specific TGF-*β* receptor kinases inhibitor. SB431542 is a selective small molecular antagonist of TGF-*β* receptor kinases that specifically binds to the ATP binding domains of the activating receptor-like kinase receptors, ALK5 (the TGF-*β* type I receptor), and other TGF-*β* family receptors, including ALK4 (activing type I receptor) and ALK7 (nodal type I receptor), resulting in the inhibition of the activation of Smad 2/3 and downstream signalling driven by TGF-*β* [57]. SB431542 has been successfully utilised by several studies to block TGF-*β* driven EMT in HCC [58,59,60]. Our study also confirmed that SB431542 can significantly inhibit TGF-*β*1-mediated EMT in both Hep3B and PLC/PRF/5 cells through changes in the expression of EMT markers and reduced migratory capability. Furthermore, we have demonstrated that SB431542 effectively decreased TGF-*β*1-induced expression of PD-L1. We have previously shown that reversal of EMT can reduce EMT induced expression of immune checkpoints such as PD-L1 [33]. To confirm the link between TGF-*β*1-induced EMT and PD-L1 expression in HCC, we utilised siRNA to knockdown the expression of PD-L1 in HCC cells. The loss of PD-L1 expression resulted in the inhibition of TGF-*β*1-mediated EMT monitored by reversal of EMT marker expression along with reduced motility. These findings provide a valuable strategy to potentially enhance the efficacy of Sorafenib by utilising a combinatorial treatment approach with the PD-L1 blocker and TGF-*β* inhibitor. However, the TGF-*β* inhibitor drug in cancer therapy can cause side effects in patients based on toxicity in preclinical studies [61,62,63]. As TGF-*β* inhibitors are not potent cytotoxic compounds, the side-effects following treatment can be minimised or avoided with adjustments such as dose assessment, selection of appropriate therapeutic combination with either chemotherapy or radiotherapy or immunotherapy, and using robust biomarkers to select appropriate patients who can benefit from the treatment [62,64,65,66]. For an instance, Galunisertib, a TGF-*β* inhibitor drug, caused cardiac toxicity in preclinical animal models [61,65]. However, the development of an intermittent dosing schedule by pharmacological, pharmacodynamics, and toxicology modelling was utilised in glioma patients, leading to no cardiac toxicity [65,67,68,69]. Not much is known regarding the possible toxicity of SB431542 in patients with HCC, and thus further studies to ascertain the adverse effects of the combination treatment involving the TGF-*β* inhibitor including SB431542 and means to reduce these adverse effects in HCC experimental models is warranted.

Ours is the first study to report combined inhibition of the TGF-*β*1-induced EMT and immune checkpoint to circumvent resistance to Sorafenib in HCC. We demonstrated a synergistic effect of Sorafenib and SB431542 as a combination drug treatment with the web-based application SynergyFinder 2.0. We reported enhanced cytotoxicity of Sorafenib against HCC cells when combined with EMT inhibitor and PD-L1 inhibition compared with Sorafenib alone, as demonstrated by the cell viability assay. The IC_50_ value of Sorafenib was significantly reduced with the combination approach. The combination treatment approach also resulted in the reversal of the EMT phenotype along with a trend in reduction of the migratory ability of HCC cells.

Previous studies have exploited numerous Sorafenib resistance mechanisms in HCC to develop an effective combination approach to treat Sorafenib refractory HCC. Studies have reported the use of the anti-epileptic drug Valproic acid (VPA) to overcome resistance to Sorafenib in HCC [70,71,72]. Treatment with Ulinastatin, a urinary trypsin inhibitor, inhibited TNF-*α* to overcome resistance to Sorafenib in HCC cells [43]. Another study revealed that the inhibition of macrophage-derived chemokine CCL22 with C-021 when combined with Sorafenib increased anti-tumour efficacy in vivo [7]. Furthermore, depletion of long non-coding RNA H19 improved Sorafenib sensitivity in HCC cells by decreasing the expression of miR-675, resulting in the inhibition of EMT [73]. Our findings also revealed that TGF-*β*1-induced EMT may be involved in upregulated immune checkpoint expression, thus resulting in the development of Sorafenib resistance in HCC cells.

## 5. Conclusions

The combined targeting of EMT and PD-L1 can be an efficacious approach to circumvent Sorafenib resistance in HCC. Validation of our in-vitro findings in in-vivo settings to assess whether combination therapy is accompanied by changes in tumor growth and metastatic capabilities is warranted. The encouraging results presented herein warrant future studies for the application of TGF-*β* and immune checkpoint inhibitors in combination with Sorafenib to treat Sorafenib reticent HCC.

## Figures and Tables

**Figure 1 jcm-10-01889-f001:**
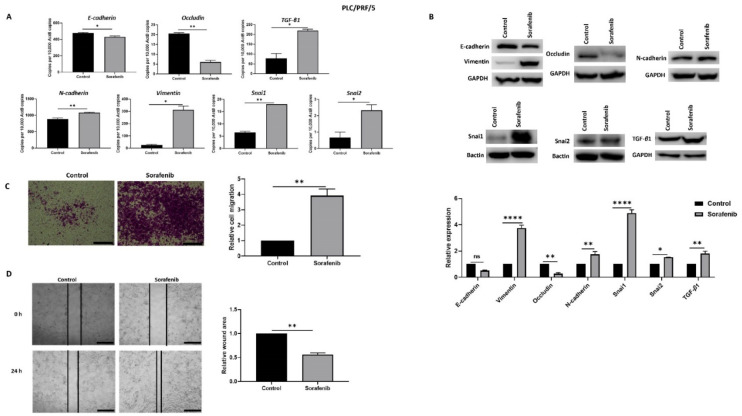
PLC/PRF/5 cells undergo EMT after Sorafenib treatment. (**A**) qRT-PCR revealed decrease in E-cadherin and Occludin and increase in TGF-β1, N-cadherin, Vimentin, Snai1, and Snai2 after treatment with 6.965 µM Sorafenib for 72 h. (**B**) Western blot analysis showed downregulation of E-cadherin and Occludin and upregulation of Vimentin, N-cadherin, Snai1, Snai2, and TGF-β1 upon treatment with Sorafenib. GAPDH and Bactin were used as loading controls. (**C**) The migratory capability of PLC/PRF/5 cells was enhanced upon treatment with Sorafenib as revealed by the transwell migration assay (scale bar = 500 µm). The number of motile cells was directly proportional to the absorbance of Crystal Violet staining. (**D**) Wound healing assay (scale bar = 500 µm) showed higher migratory capacity of Sorafenib treated PLC/PRF/5 cells. Quantitative analysis of the wound area after 24 h Sorafenib treatment relative to the starting wound area at 0 h (*n* = 3, * *p* < 0.05, ** *p* < 0.01, **** *p* < 0.001).

**Figure 2 jcm-10-01889-f002:**
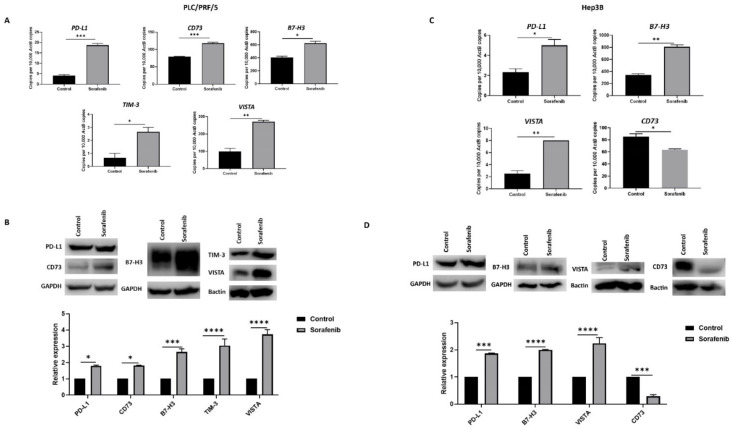
Sorafenib treatment induces immune checkpoint expression in HCC cells. (**A**) qRT-PCR demonstrated upregulation of immune checkpoint molecules PD-L1, CD73, B7-H3, VISTA, and TIM-3 in PLC/PRF/5 cells upon treatment with Sorafenib. (**B**) Western blot analysis revealed increased expression of PD-L1, CD73, B7-H3, VISTA, and TIM-3 in PLC/PRF/5 cells upon treatment with Sorafenib. GAPDH and Bactin were used as loading control. (**C**) qRT-PCR demonstrated upregulation of immune checkpoint molecules PD-L1, B7-H3, and VISTA and downregulation of CD73 in Hep3B cells treated with Sorafenib. (**D**) Western blot analysis revealed increased expression of PD-L1, B7-H3, and VISTA and decreased expression of CD73 with minimal expression of TIM-3 in Hep3B cells post Sorafenib treatment. GAPDH and Bactin were used as loading control (*n* = 3, * *p* < 0.05, ** *p* < 0.01, *** *p* < 0.005, **** *p* < 0.001).

**Figure 3 jcm-10-01889-f003:**
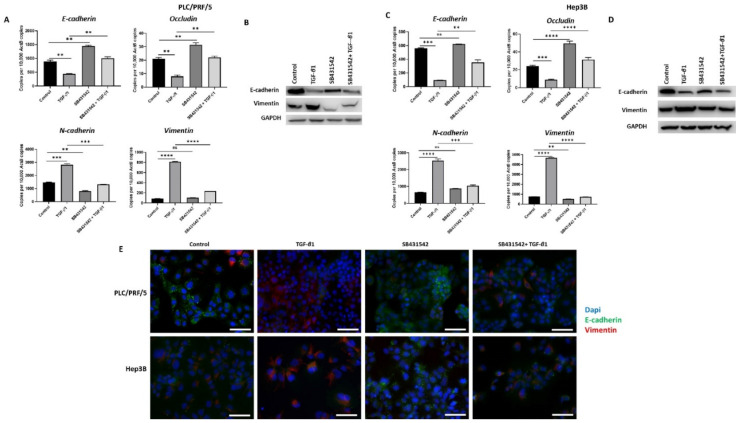
SB431542 inhibits TGF-β1-mediated EMT in HCC cells. (**A**) qRT-PCR revealed an increase in E-cadherin and Occludin and a decrease in N-cadherin and Vimentin expression and (**B**) western blot showed an increase in E-cadherin and reduction in Vimentin following TGF-β1 and SB431542 treatments compared with cells treated with TGF-β1 alone in PLC/PRF/5 cells. GAPDH was used as the loading control. (**C**) qRT-PCR revealed increased expression of E-cadherin and Occludin and lower expression of N-cadherin and Vimentin and (**D**) western blot analysis demonstrated higher expression of E-cadherin and lower expression of Vimentin following treatment with TGF-β1 and SB431542 in Hep3B cells. GAPDH was used as the loading control. (**E**) Fluorescence microscopy demonstrated upregulation of E-cadherin and downregulation of Vimentin in cells treated with TGF-β1 and SB431542 compared with cells treated with TGF-β1 alone in both PLC/PRF/5 and Hep3B cells (scale bar = 200 µm) (*n* = 3, ** *p* < 0.01, *** *p* < 0.005, **** *p* < 0.001, ns: not significant).

**Figure 4 jcm-10-01889-f004:**
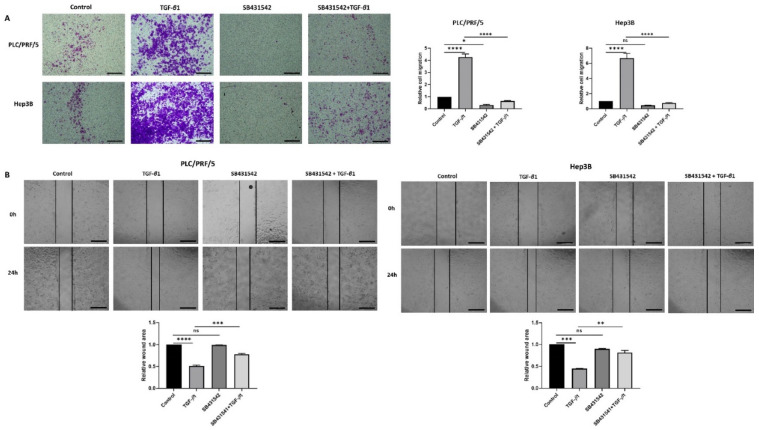
SB431542 reverses migratory capability of TGF-β1-stimulated HCC cells. (**A**) Transwell migration assay revealed reduced migratory capability of PLC/PRF/5 and Hep3B cell lines upon treatment with SB431542 despite stimulation with TGF-β1 (scale bar = 500 µm). Quantitative analysis of motile cells was determined by measuring the absorbance of Crystal Violet staining. (**B**) Wound healing assay confirmed decreased motility of PLC/PRF/5 and Hep3B cells upon treatment with SB431542 despite stimulation with TGF-β1 (scale bar = 500 µm). Wound area was analysed with TGF-β1, SB431542, or combination treatment for 24 h relative to the starting wound area at 0 h (*n* = 3, * *p* < 0.05, ** *p* < 0.01, *** *p* < 0.005, **** *p* < 0.001).

**Figure 5 jcm-10-01889-f005:**
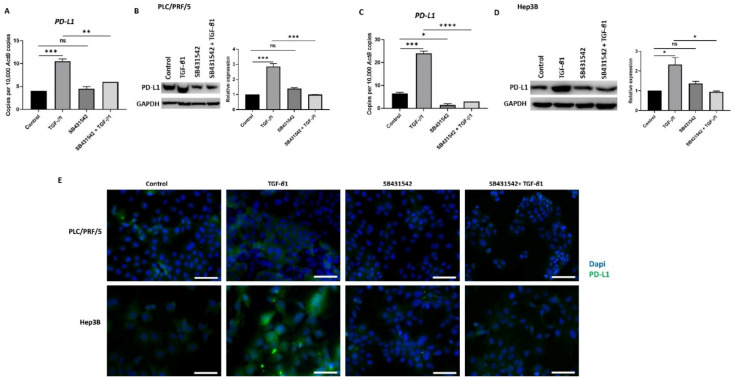
SB431542 inhibits TGF-β1-induced expression of PD-L1 in HCC cells. (**A**) qRT-PCR and (**B**) western blot analysis revealed reduced expression of PD-L1 in PLC/PRF/5 cells after treatment with TGF-β1 and SB431542. GAPDH was used as loading. (**C**) qRT-PCR and (**D**) western blot analysis revealed reduced expression of PD-L1 following SB431542 treatment in Hep3B cells. GAPDH was used as loading control. (**E**) Fluorescence microscopy revealed reduced expression of PD-L1 following SB431542 treatment in PLC/PRF/5 and Hep3B cells exposed to TGF-β1 (scale bar = 200 µm) (*n* = 3, * *p* < 0.05, ** *p* < 0.01, *** *p* < 0.005, **** *p* < 0.001).

**Figure 6 jcm-10-01889-f006:**
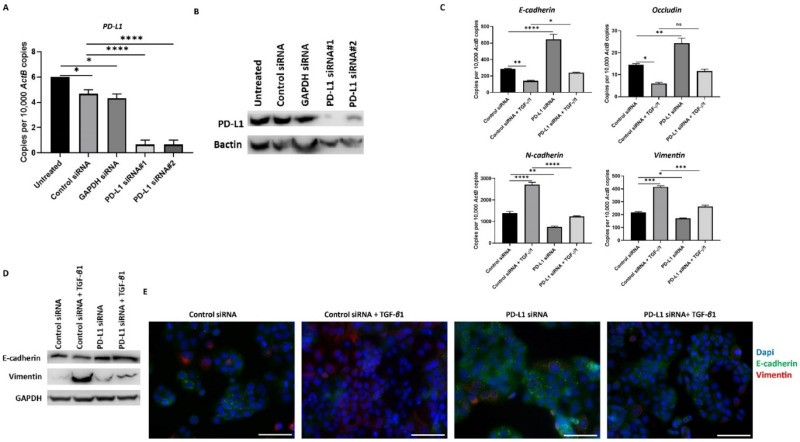
Silencing of PD-L1 reverses TGF-β1-mediated EMT in PLC/PRF/5 cells. Transfection of HCC cells with two specific PD-L1 siRNA effectively knockdown PD-L1 expression, as demonstrated by (**A**) qRT-PCR and (**B**) western blot analysis. Bactin was utilised as the loading control. (**C**) qRT-PCR demonstrated that silencing of PD-L1 resulted in upregulation of E-cadherin and Occludin along with downregulation of N-cadherin and Vimentin expression in PLC/PRF/5 cells. (**D**) Western blot analysis showed increase in E-cadherin and decrease in Vimentin upon knockdown of PD-L1 in PLC/PRF/5 cells. GAPDH was utilised as loading control in western blot analysis. (**E**) Fluorescence microscopy showed elevation of E-cadherin and decline of Vimentin following PD-L1 knockdown in PLC/PRF/5 cells (scale bar = 200 µm) (*n* = 3, * *p* < 0.05, ** *p* < 0.01, *** *p* < 0.005, **** *p* < 0.001).

**Figure 7 jcm-10-01889-f007:**
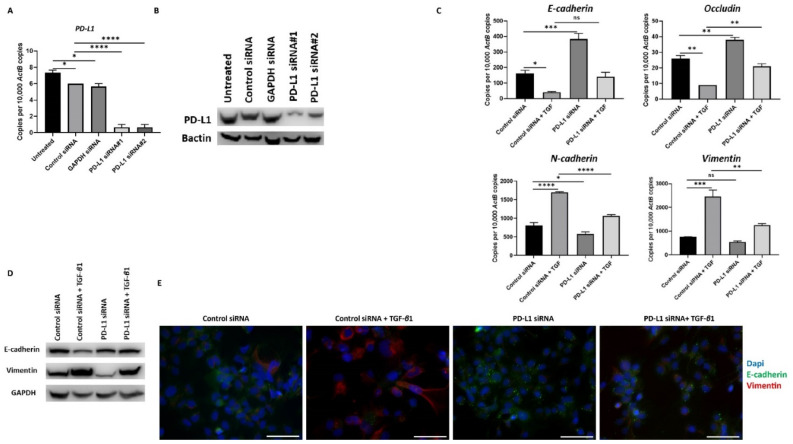
Silencing of PD-L1 reverses TGF-β1-mediated EMT in Hep3B cells. (**A**,**B**) Transfection of Hep3B cells with two specific PD-L1 siRNA effectively knockdown PD-L1 expression as demonstrated by qRT-PCR and western blot analysis. Bactin was used as the loading control. Silencing of PD-L1 resulted in increased E-cadherin and Occludin expression along with reduction in N-cadherin and Vimentin expression in Hep3B cells as assessed by (**C**) qRT-PCR and (**D**) western blot analysis. GAPDH was utilised as the loading control. (**E**) Fluorescence microscopy showed an increase in E-cadherin and a decrease in Vimentin following PD-L1 knockdown in Hep3B cells (scale bar = 200 µm). (*n* = 3, * *p* < 0.05, ** *p* < 0.01, *** *p* < 0.005, **** *p* < 0.001).

**Figure 8 jcm-10-01889-f008:**
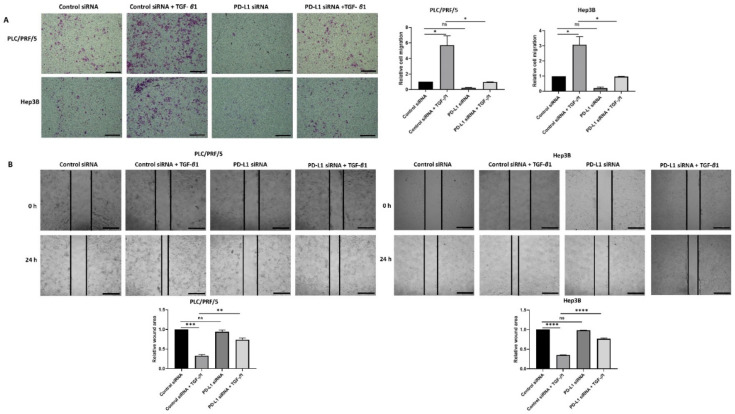
Silencing of PD-L1 reverses migratory ability of TGF-β1-stimulated HCC cells. (**A**) Transwell migration assay revealed reduced migratory capability of PLC/PRF/5 and Hep3B cells following PD-L1 knockdown despite stimulation with TGF-β1 (scale bar = 500 µm). The number of motile cells was determined by measuring the absorbance of Crystal Violet staining. (**B**) Decreased motility of PLC/PRF/5 and Hep3B cells following PD-L1 knockdown despite stimulation with TGF-β1 was validated by the wound healing assay (scale bar = 500 µm). The analysis of the wound area after 24 h treatment with siRNA alone or when combined with TGF-β1 relative to the starting wound area at 0 h (*n* = 3, * *p* < 0.05, ** *p* < 0.01, *** *p* < 0.005, **** *p* < 0.001).

**Figure 9 jcm-10-01889-f009:**
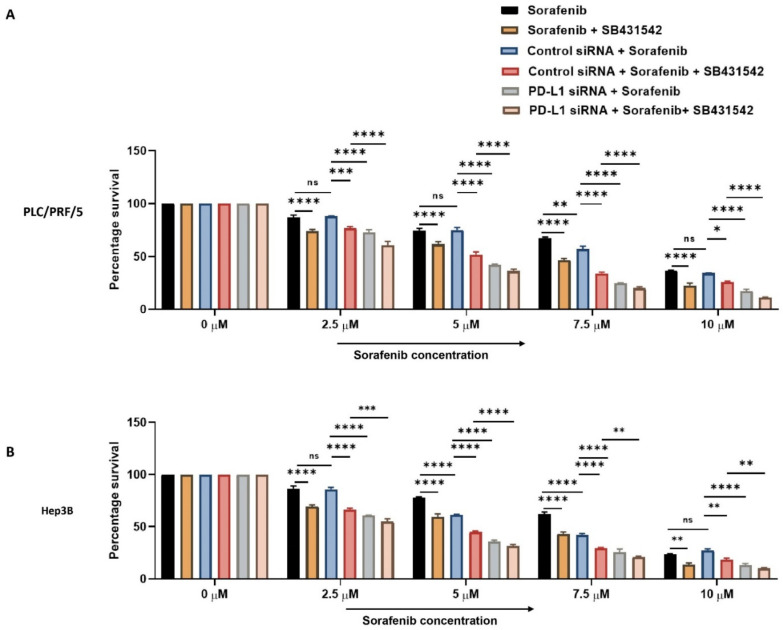
Combination treatment of PD-L1 knockdown with SB431542 can overcome Sorafenib resistance in HCC cells. Cell viability assay revealed that percentage survival of cells significantly reduced with the combination of increasing concentrations of Sorafenib with knockdown of PD-L1 and SB431542 compared to Sorafenib alone or Sorafenib with either SB431542 or PD-L1 knockdown or control siRNA treatment (**A**) PLC/PRF/5 and (**B**) Hep3B cells (*n* = 3, * *p* < 0.05, ** *p* < 0.01, *** *p* < 0.005, **** *p* < 0.001).

**Figure 10 jcm-10-01889-f010:**
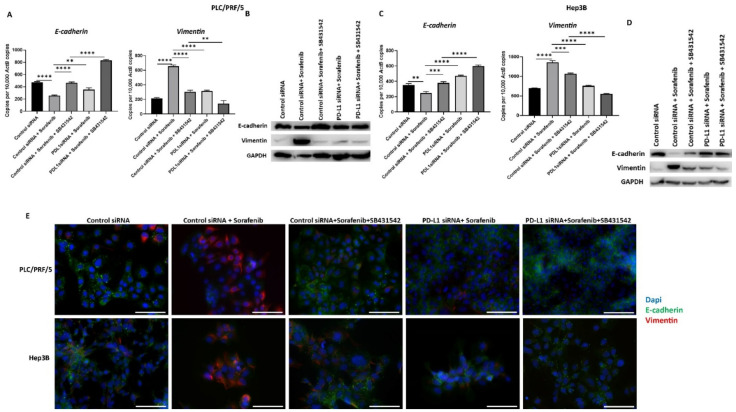
Combination treatment of Sorafenib with PD-L1 knockdown and SB431542 can reverse Sorafenib-driven EMT in HCC cells. Combination treatment of Sorafenib with PD-L1 knockdown and SB431542 elevated E-cadherin and downregulated Vimentin in PLC/PRF/5 cells, as demonstrated by (**A**) qRT-PCR and (**B**) western blot analysis. GAPDH was used as the loading control. Similar combination treatment of Sorafenib with PD-L1 knockdown and SB431542 resulted in upregulation of E-cadherin and downregulation of Vimentin in Hep3B cells, as demonstrated by (**C**) qRT-PCR and (**D**) western blot analysis. GAPDH was used as the loading control. (**E**) Fluorescence microscopy revealed upregulation of E-cadherin and downregulation of Vimentin in both PLC/PRF/5 and Hep3B cells following combination treatment of Sorafenib with PD-L1 knockdown and SB431542 (scale bar = 200 µm) (*n* = 3, ** *p* < 0.01, *** *p* < 0.005, **** *p* < 0.001).

**Figure 11 jcm-10-01889-f011:**
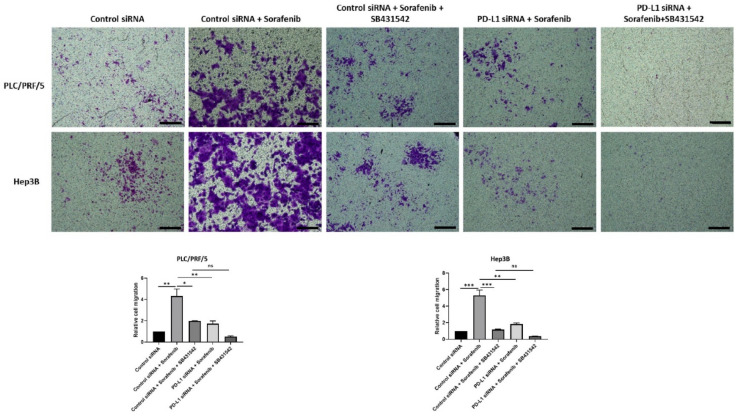
Combination treatment of Sorafenib with PD-L1 knockdown and SB431542 can reverse Sorafenib-induced migration in HCC cells. Migratory ability of PLC/PRF/5 and Hep3B cells was attenuated by combination treatment of Sorafenib with PD-L1 knockdown and SB431542, as revealed by images of the transwell migration assay (scale bar = 500 µm). The number of motile cells was determined by measuring the absorbance of Crystal Violet staining (*n* = 3, * *p* < 0.05, ** *p* < 0.01, *** *p* < 0.005).

**Table 1 jcm-10-01889-t001:** List of primers for quantitative reverse transcription-PCR.

Primers	Sequence (5′-3′)
*Snai1* forward	GCTGCAGGACTCTAATCCAGA
*Snai1* reverse	ATCTCCGGAGGTGGGATG
*TGF-β1* forward	TACCTGAACCCGTGTTGCTCTC
*TGF-β1* reverse	GTTGCTGAGGTATCGCCAGGAA
*TNF-α* forward	CCCAGGGACCTCTCTCTAATC
*TNF-α* reverse	TCTCAGCTCCACGCCATT
*TIM-3* reverse	GACTCTAGCAGACAGTGGGATC
*TIM-3* reverse	GGTGGTAAGCATCCTTGGAAAGG

**Table 2 jcm-10-01889-t002:** List of antibodies.

Antibodies	Cat. No.	Manufacturer	Antibody Category	Western Blot	Immunofluorescence
TGF-β1	MA5-15065	Thermofisher	Primary	1:1000	
Ki67	ab66155	Abcam	Primary		1:200
β-Actin	4967s	Cell Signaling	Primary	1:2000	
Snai1	3879S	Cell Signaling	Primary	1:1000	
Snai2	9585S	Cell Signaling	Primary	1:1000	
VISTA	ab235362	Abcam	Primary	1:1000	
TIM-3	ab241332	Abcam	Primary	1:1000	

## Data Availability

The data presented in this study are available on request from the corresponding author.

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
