# Peer review of "Combined Inhibition of TGF-β1-Induced EMT and PD-L1 Silencing Re-Sensitizes Hepatocellular Carcinoma to Sorafenib Treatment"

_jcm, 2021, doi:10.3390/jcm10091889_

Round 1

Reviewer 1 Report

Tis manuscript has been improved regardin to the two reviewings.

Author Response

Thank you for the consideration.

Reviewer 2 Report

This is a revised version of the manuscript “Combined Inhibition of TGF-β1-induced EMT and PD-L1 Silencing Re-sensitizes Hepatocellular Carcinoma to Sorafenib Treatment”. The authors significantly improved the quality and clarity of the data. However, there are several recommendations and concerns regarding data integrity.

  1. The authors may inform a full-term before using an abbreviation in the main manuscript. EMT on page 3.
  2. The name of control material is missing in Figure 1B. And, the Western blotting figures and quantification data should be in the same order (i.e., TGF-β1 data is present in the middle of the WB data, but the quantification is the last).
  3. Western blotting figures and quantification data should be presented in the same order in Suppl. Figure 2B.
  4. Please match the order of data presentation between Figures 2A and 2B.
  5. Western blotting figures and quantification data should be in the same order in Figure 2B.
  6. In the results, the authors mentioned their other manuscripts which are either under review or revision. Please make sure these manuscripts are not the same as the others. If they are the same, the authors may need to match the status.
  7. Was “treatment” a correct term in the subtitle on page 15?
  8. Authors showed the silencing of PD-L1 on EMT data with PLC/PRF/5 cells in Figure 6, and then presented migratory data with PLC/PRF/5 AND Hep3B cells in Figure 7. The EMT data with Hep3B cells were in Suppl. Figure 4. It will be better to show both data in the main manuscript unless the authors have a different intent.
  9. Figure 8 was still too busy. Is it necessary to show all doses?
  10. Authors showed that PD-L1 siRNA was superior to control siRNA in terms of silencing PD-L1 and EMT reversal, but there was no difference as the combination treatment with sorafenib + SB431542 using both cell lines in Figure 10. Please make sure that your data interpretation was correct.

Author Response

A point-by-point response to the reviewer's comments have been provided in the attached file.
